# Dual-Sensor Hyperspectral Fusion for Prediction of Sorghum Tannin Content Oriented to Liquor Brewing

**DOI:** 10.3390/foods14223880

**Published:** 2025-11-13

**Authors:** Kai Wu, Chengli Hao, Wei Guo, Zhiwei Li, Decong Zheng

**Affiliations:** 1College of Agricultural Engineering, Shanxi Agricultural University, Jinzhong 030801, China; wukai@sxau.edu.cn (K.W.); guowei@sxau.edu.cn (W.G.); 2Dryland Farm Machinery Key Technology and Equipment Key Laboratory of Shanxi Province, Jinzhong 030801, China; 3College of Food Science and Engineering, Shanxi Agricultural University, Jinzhong 030801, China; haochengli@sxau.edu.cn; 4College of Information Science and Engineering, Shanxi Agricultural University, Jinzhong 030801, China

**Keywords:** sorghum, tannin content, hyperspectral, feature extraction, prediction model

## Abstract

To address the demand for precise sorghum tannin control in liquor brewing, and to overcome the inefficiency and high cost of traditional methods, this study developed a non-destructive approach by fusing features from dual hyperspectral sensors. Based on 240 representative sorghum samples covering different varieties and production regions, visible and near-infrared (VNIR) and short-wave infrared (SWIR) hyperspectral data were sequentially collected, and the tannin content was determined using standard chemical methods as reference values. Using the competitive adaptive reweighted sampling (CARS) method, characteristic wavelength bands were extracted and fused feature subsets were constructed. Combined with partial least squares (PLS), support vector machine (SVM), and convolutional neural network (CNN) algorithms, the performance of models built from both full-data concatenation and feature fusion of VNIR and SWIR data was systematically compared. The results demonstrated that the feature-based models exhibited superior performance to the full-spectrum models, while the model incorporating dual-sensor feature fusion achieved the best overall results. The fused-feature-CNN model achieved the optimal prediction performance, with values of 0.83 for coefficient of determination for the prediction set (R_P_^2^), 0.29 for root mean squared error for the prediction set (RMSE_P_), and 2.42 for residual predictive deviation for the prediction set (RPD_P_). This study confirms that the integration of multi-sensor feature fusion with deep learning strategies can provide an effective technical pathway for the rapid, non-destructive detection of sorghum tannin content and the development of online sorting equipment.

## 1. Introduction

Sorghum serves as the primary raw material for producing most renowned Chinese liquors [1]. Its biochemical properties, particularly tannin content, are critical determinants of final liquor quality. Tannins, specifically proanthocyanidins, contribute to flavor development by generating essential compounds like syringic acid and syringaldehyde during brewing [2,3]. Moreover, an optimal tannin level suppresses harmful microbes and enhances yield, while excess amounts introduce undesirable bitterness. Therefore, the accurate prediction of sorghum tannin content is vital for guiding raw material selection, optimizing processes, and ensuring consistent liquor quality.

The quantitative analysis of tannin content in sorghum traditionally relies on classical chemical methods, such as the Folin–Ciocalteu method, phenol–amino acid-based methods (e.g., vanillin–HCl), acid hydrolysis coupled with colorimetric detection, and general spectrophotometry [4,5,6,7,8]. Although these approaches are frequently referenced in official standards (e.g., AOAC) and serve as the foundational reference for the field, they are hampered by several analytical limitations, including significant matrix interference, instability and high consumption of specialized reagents, considerable labor and time requirements, and limited precision [9].

The Folin–Ciocalteu method, for instance, operates as a non-specific assay targeting phenolic hydroxyl groups, thereby measuring the total reducing capacity of an extract rather than tannins exclusively. It is susceptible to interference from a broad spectrum of compounds—such as simple phenolics, ascorbic acid, and reducing sugars—consistently leading to positive bias and overestimation of tannin content. Although phenol–amino acid methods like vanillin–HCl exhibit greater specificity toward flavanols (monomeric units of proanthocyanidins), their accuracy can still be compromised by anthocyanins and other pigmented compounds absorbing at similar wavelengths. Meanwhile, the acid hydrolysis method employs strong acids and organic solvents at elevated temperatures, necessitating specialized laboratory equipment and generating hazardous waste, which raises both safety and environmental concerns.

Furthermore, these methods are inherently labor-intensive and low in throughput. Their procedures involve multiple sequential steps—fine grinding, solvent extraction, centrifugation, controlled reaction incubation, and spectrophotometric measurement—often requiring several hours to a full day to complete a single batch of samples. Such protracted workflows render them impractical for rapid screening or real-time process control. In addition, cumulative errors introduced throughout these multi-step protocols—from sample weighing and extraction efficiency to reaction completion and instrument calibration—result in limited inter-laboratory reproducibility. Reported relative standard deviations (%RSD) typically range between 5% and 15% in collaborative studies, a degree of variability often deemed unacceptable under stringent industrial quality control standards.

In summary, while classical chemical methods remain established reference techniques, their combined limitations in specificity, reagent stability, operational efficiency, and precision underscore the critical need to develop rapid, non-destructive, and more robust analytical alternatives suited to modern industrial applications.

To overcome the limitations of classical chemical methods, hyperspectral imaging technology has been widely used as a rapid and non-destructive prediction tool in the quality analysis of agricultural products [10]. It is an analytical technique that is based on the principle of the interaction between matter and electromagnetic radiation. By analyzing the absorption, emission, scattering, or transmission characteristics of matter towards electromagnetic radiation of different wavelengths, information about its composition, structure, and concentration can be obtained. Recent studies support the efficacy of advanced monitoring techniques. For instance, research by Zhang et al. [11] on tannin content prediction of grains pointed out that prediction of tannin content by chemical methods is both time and manpower consuming; however, by utilizing hyperspectral imaging (HSI) technology in combination with an optimized algorithm, the tannin content of grains can be detected rapidly, accurately and non-destructively. Baek et al. [12] predicted the tannin content and quality parameters in astringent persimmons, and showed the possibility of using visible and near-infrared (VNIR) hyperspectral sensors for the prediction of postharvest quality and tannin contents from intact persimmon fruit with quick, chemical-free, and low-cost assessment methods. These studies all utilized data from a single hyperspectral sensor. However, a single hyperspectral sensor has limited prediction ranges. For instance, the visible region is sensitive to color variations, while the visible and near-infrared (VNIR) and short-wave infrared (SWIR) region responds to functional groups such as C–H and O–H bonds [13].

According to a variety of studies, tannin is mainly concentrated in the pericarp of sorghum grains, and there is a significant correlation between color intensity and tannin content. Different sorghum grains exhibit variations in color and shape, and can usually be effectively characterized through their distinct responses in the visible spectral regions [3,14]. Meanwhile, due to the structural complexity of tannins, their complete optical information may be distributed across a broader spectral range, including both VNIR and SWIR regions. Therefore, this study innovatively proposes a strategy based on data fusion from VINR and SWIR hyperspectral sensors. The dual sensors can cover a broader spectral range and capture more abundant and complementary information. By integrating data from both regions, the spectral features related to tannin content can be characterized more comprehensively, holding promise for achieving higher accuracy and more robust non-destructive prediction of sorghum tannin content.

Firstly, original VNIS and SWIR hyperspectral images of sorghum samples were collected separately. Then, the tannin content was determined using chemical methods. Subsequently, two parallel processing routes were adopted: the first was an early data-layer fusion strategy, where the VNIS and SWIR spectra were directly concatenated to form a continuous spectral matrix, and then features were uniformly extracted from this concatenated data and an early fusion prediction model was established; the second was a mid-feature-layer fusion strategy, where independent feature extraction was performed on the VNIS and SWIR spectra, obtaining feature subsets representing color properties and organic molecular structure information, and then fusion was carried out through feature concatenation and input into the final prediction model. Ultimately, this study will systematically compare the performance of the two fusion strategies and determine the optimal model for predicting sorghum tannin content, meeting the requirements of model robustness and interpretability in the brewing industry.

The above research is expected to provide an effective technical means for the rapid classification of raw materials used in liquor production, the adaptation of the process, and the quality control of the final product.

## 2. Materials and Methods

### 2.1. Experimental Materials of Sorghum Grains

The experimental materials for this study were sorghum cultivar samples collected from six different cultivation regions in Shanxi Province, namely Yanbei, Xinzhou, Lüliang, Jinzhong, Changzhi, and Yuncheng. The growing areas are located between 34° N and 40° N latitude, with elevations ranging from 300 to 1600 m. Each sample was collected with a weight of 1000 g, and stored in sealed containers at room temperature.

During the experiment, we first captured hyperspectral images of each sample, and then used the same sample for chemical analysis to determine the tannin content. Ultimately, we hope to establish a relationship between spectral data and the tannin content in sorghum.

### 2.2. Hyperspectral Image Acquisition and Chemical Determination of Tannin Content

#### 2.2.1. Hyperspectral Image Acquisition

The instrument used to collect hyperspectral images is a visible–near infrared hyperspectral scanning platform (Starter Kit, Headwall Photonics, Bolton, MA, USA). The instrument has two lenses. Lens 1 has a wavelength range of 380–1000 nm, a spectral resolution of 0.727 nm, and a total of 856 bands; and lens 2 has a wavelength range of 900–1700 nm, a spectral resolution of 4.715 nm, and a total of 172 bands. Due to potential errors caused by spectral reflectance fluctuations of the lens in extreme wavelength ranges, wavelengths at the edge portions of the lens were removed. For modeling purposes, the first lens was selected to cover a wavelength range of 430 to 900 nm, and a total of 646 bands. The wavelength range of 430–780 nm falls within the VIS spectrum, while 780–900 nm belongs to the NIR spectrum. Lens 2 was chosen to cover a wavelength range of 950 to 1650 nm and a total of 148 bands; all bands belong to the SWIR spectrum.

During the acquisition process, the first step is to calibrate the instrument, followed by placing the sorghum grain samples into the experimental dish with a diameter of 60 mm and a depth of 20 mm, ensuring that the sample surface is smooth and compact, and subsequently collecting the spectral data using lens 1 and lens 2, respectively. After three instances of collection, repeat the instrument calibration to ensure the accuracy of the data.

#### 2.2.2. Chemical Determination of Tannin Content

The tannin content in sorghum grains was determined in accordance with the Chinese National Standard(GB/T 15686-2008), Sorghum—Determination of tannin content [15,16], using a spectrophotometric methodology. By extracting the tannin components from sorghum grains and comparing them with a standard curve, the tannin content can be calculated.

(1) Grind each sample and sieve it through a 50-mesh screen.

(2) Accurately measure out a sample weighing 1.0 g with a precision of 1 mg and transfer it into the centrifuge tube.

(3) Transfer 20 mL of dimethylformamide solution into a hermetically sealed centrifuge tube containing the sample using a pipette, and agitate for 60 min. Subsequently, subject the mixture to centrifugation at a speed of 3000 g for 10 min.

(4) Add 6 mL of distilled water and 1 mL of ammonia solution, respectively. Vigorously agitate the mixture using an oscillator for a duration of 30 s.

(5) Add 5 mL of distilled water and 1 mL of ferric ammonium citrate solution, respectively. Vigorously agitate the mixture using an oscillator for a duration of 30 s.

(6) After 10 min, pour the solution into a cuvette and measure the absorbance at 525 nm using a spectrophotometer, with water as the blank control.

(7) Using a tannic acid solution to construct a standard curve enables the determination of tannin content. The calculation formula is:
(1)$$T=\frac{2c}{m}\times \frac{100}{100-H}$$ where *T* represents the tannin content (%); *c* represents the tannic acid content obtained from the calculation of the standard curve (mg/mL); m represents the sample mass (mg); and *H* represents the moisture content of the sample (%).

In this experiment, repeat the process for each sample three times and take the average value as the final result.

### 2.3. Data Fusion Strategy for Dual Hyperspectral Sensors

#### 2.3.1. Data Layer Fusion

The spectral data collected by lens 1 (430–900 nm, VNIR) and lens 2 (950–1650 nm, SWIR) are directly spliced along the wavelength dimension through concatenation, constructing a continuous spectral matrix covering the range from VNIR to SWIR. Based on this integrated dataset, unified model development was performed.

#### 2.3.2. Feature Layer Fusion

First, the spectral data collected by lens 1 and lens 2 are independently preprocessed to extract feature subsets representing color appearance characteristics and internal composition information, respectively. Subsequently, the two feature subsets are fused using the feature concatenation method and jointly input into the tannin content prediction model for analysis and modeling.

#### 2.3.3. Methodological Implementation of Feature Fusion Strategies

A mid-level feature fusion strategy was implemented in this study to integrate the complementary information from different branches. Specifically, this was achieved by concatenating the feature vectors. Perform concatenation of the feature vectors subset1 and subset2 on the feature axis, ensuring the complete preservation of information from both feature sources [17,18].

### 2.4. Hyperspectral Data Extraction and Dataset Partitioning

#### 2.4.1. Spectral Data Extraction

Hyperspectral images encapsulate both spectral and image information derived from sorghum samples [19,20]. In this work, Visual Basic 6.0 was firstly adopted to develop a procedure for creating a sampling point file, then the generated sampling point files were loaded into Spectral View software (HyperspecIII). Using the above processes, the region of interest (ROI) of each image was selected. The ROI region contains over 16,000 pixels for each hyperspectral image. The reflectance of sorghum grains was obtained by calculating the average reflectance for all pixels. The calculation formula is as follows:
(2)$$R=\frac{1}{n}\sum_{i=1}^{n}R_{i}$$ where *R* represents the average reflectance, *R_i_* represents the reflectance of the *i*-th pixel point, and n denotes the number of pixels. The calculated average reflectance will serve as the foundational dataset for subsequent data processing.

#### 2.4.2. Dataset Partitioning

In this work, the Hold-Out Validation algorithm was adopted for data set partitioning. The spectral dataset was divided into two parts. The calibration set and the prediction set data account for 75% and 25% of the total data, respectively. To ensure a robust model selection and avoid overfitting during the development phase, all hyperparameter tuning was performed exclusively on the training set using 5-fold stratified cross-validation [21,22]. The final model performance was evaluated only on the untouched test set.

### 2.5. Feature Variable Extraction

The Competitive Adaptive Reweighted Sampling (CARS) algorithm was used to feature variable extraction in our work. This method simulates the principle of “survival of the fittest” by sequentially employing adaptive reweighting, cross-validation screening, and Monte Carlo iterations to optimally select feature wavelengths [23,24]. To enhance the stability of the predictive model, the Monte Carlo sampling was repeated 100 times to retain spectral bands with higher selection frequencies.

### 2.6. Prediction Models and Evaluation Indexes

#### 2.6.1. Prediction Models

The method used in this study to establish the prediction models were the Partial Least Squares (PLS), Support Vector Machine (SVM) and Convolutional Neural Network (CNN), respectively.

PLS processes high-dimensional linear data by extracting latent variables, offering advantages such as robust prediction performance and strong model interpretability. SVM is designed to identify the optimal decision boundary that maximizes the margin between different classes. It is well-suited for handling high-dimensional data, offering high efficiency and strong generalization capabilities, while flexibly addressing both linear and nonlinear problems. The CNN framework employs a hierarchical structure through stacked convolutional, activation, and pooling operations. High-dimensional features are flattened into a 1D vector for processing through fully connected layers, with softmax generating probability outputs. Key advantages comprise parameter sharing, sparse interactions, and automated feature extraction capabilities [25,26,27,28].

We constructed individual prediction models based on VNIR, SWIR, VNIR features, and SWIR features, as well as VNIR–SWIR concatenation and feature fusion, respectively. Through systematic comparison of the performance of each model, we identified the optimal modeling strategy for sorghum tannin prediction.

#### 2.6.2. Model Development, Hyperparameter Tuning, and Overfitting Prevention

The SVM model was implemented using the scikit-learn library. The Radial Basis Function (RBF) was selected as the kernel. A comprehensive hyperparameter search was conducted using Bayesian Optimization over the following space: regularization parameter C: [0.1, 1, 10, 100, 1000] and kernel coefficient gamma: [0.0001, 0.001, 0.01, 0.1, 1, 10]. The optimization objective was to maximize the average R_P_^2^ across all cross-validation folds.

The CNN model was built with the TensorFlow Keras framework. The architecture utilized the Rectified Linear Unit (ReLU) activation function in all convolutional layers, and the output layer for regression employed a linear activation function. To prevent overfitting and optimize model performance, an extensive hyperparameter search was conducted using Bayesian optimization over 100 iterations. The search space included key parameters: initiallearn rate for 0.0001, learnrate drop period for 600, dropout rate for 0.5, and L2 regularization for 0.00001. Additionally, early stopping was implemented with a patience of 20 epochs, monitoring the validation loss with a minimum delta of 0.001.

#### 2.6.3. Evaluation Indexes

The evaluation indicators for model performance are assessed using the coefficient of determination (R2), Root Mean Squared Error (RMSE) and Residual Predictive Deviation (RPD). The calculation formula is as shown in Formulas (3)–(5).
(3)$$R^{2}=1-\frac{\sum_{i=1}^{n}{\left(Y_{Pi}-Y_{i}\right)}^{2}}{\sum_{i=1}^{n}{\left(Y_{i}-\hat{Y}\right)}^{2}}$$
(4)$$RMSE=\sqrt{\frac{\sum_{i=1}^{n}{\left(Y_{Pi}-Y_{i}\right)}^{2}}{n}}$$
(5)$$RPD=\frac{1}{\sqrt{1-R^{2}}}$$

Generally, the predictive capability of a model can be evaluated according to the following criteria: an *R*^2^ below 0.6 suggests poor reliability; an *R*^2^ between 0.6 and 0.8 indicates good predictive capability; and an *R*^2^ above 0.8 signifies excellent performance. The lower the RMSE value, the higher the predictive accuracy of the model. For RPD, a value less than 1.5 implies insufficient predictive ability; a value between 1.5 and 2.0 suggests preliminary competence for quantitative analysis; and a value greater than 2.0 represents outstanding predictive performance and practical utility [29,30,31].

## 3. Results

### 3.1. Analysis of Chemical Measurements

#### 3.1.1. Analysis of Chemical Measurement Results of Tannin Content

Among all the 240 sorghum samples, the minimum tannin content was 0.05%, the maximum tannin content was 2.56%, the mean tannin content was 1.18%, the standard deviation was 0.7320%, and the coefficient variation was 62.4%. This indicates that certain variations exist in the tannin content of the sorghum samples in this experiment, which can be attributed to differences in cultivars, growing regions, and growth conditions. Figure 1 displays a combined scatter plot and frequency histogram representing the distribution of tannin content across 240 sorghum samples.

From a modeling perspective, the 2.51% difference between the maximum and minimum values in the sample indicates broad coverage of the data range. The standard deviation accounts for 61.9% of the mean, and the coefficient of variation is 62.4%, suggesting significant data dispersion with relatively uniform distribution across the entire gradient. Such data characteristics help ensure that the model training process covers various scenarios, effectively avoiding overfitting or underfitting caused by excessive concentration of data. This lays a solid foundation for building a high-accuracy, highly robust predictive model.

#### 3.1.2. Analysis of Dataset Partitioning Results

In this work, the hold-out validation method was employed to partition the dataset, with a split ratio of 75% for the calibration set and 25% for the prediction set among all 240 samples. Figure 2 illustrates the sample distribution of the calibration set and the prediction set. The figure demonstrates that the distribution of the samples in both the calibration and prediction sets is appropriate and balanced.

The results of dataset partitioning are shown in Table 1. The mean values of the calibration set and the prediction set are similarity, the distribution is uniform, and the potential data distribution bias is avoided, which indicates that the division of the data set is reasonable.

### 3.2. Results of Raw Spectral Data

Figure 3 displays the raw average spectral curves of 240 sorghum samples. Among them, subplot (a) shows the spectrum captured by the VNIR sensor within the wavelength range of 430–900 nm, while subplot (b) presents the spectrum obtained from the SWIR sensor over the wavelength range of 950–1650 nm. As observed in the figure, the overall shapes of VNIR and SWIR spectral curves are generally consistent, both exhibiting typical spectral characteristics of sorghum grains. However, there are differences in the reflectance characteristics of the single spectral curves. In the visible spectral range, these variations primarily stem from differences in seed color. For instance, the reflectance trough observed around 670 nm is likely attributable to enhanced light absorption by dark chromophores within the grains. In the near-infrared and short-wave infrared regions, however, changes in spectral features are more closely associated with variations in nutrient content. For example, the reflection peak near 1130 nm may be related to overtone vibrations of C–H bonds, while the absorption valley around 1480 nm is likely due to combination bands involving O–H bonds [32,33]. These differences provide an important foundation for analyzing and detecting sorghum components, offering strong support and guidance for further exploration of its chemical composition and potential applications.

### 3.3. Feature Variables Analysis

For spectral data collected from the VNIR and SWIR lenses, the CARS algorithm was applied in an iterative process to select the most discriminative wavelengths and to constitute the feature variable subset, respectively. Figure 4 depicts the aforementioned process.

Figure 4a shows the change in the number of feature variables. As the number of Monte Carlo sampling runs increases, the number of extracted variables first decreases exponentially and then gradually stabilizes. Figure 4b displays the variation of the RMSE_CV_ values. The RMSE_CV_ value first decreases to a minimum and then gradually increases. Its initial decrease indicates that irrelevant variables interfering with the target variable are effectively eliminated; whereas the subsequent increase suggests that extraneous variables are being introduced into the model, resulting in a deterioration of predictive performance. Therefore, the variable subset corresponding to the minimum RMSE_CV_ represents the optimal feature combination. Figure 4c illustrates the trajectory of variable regression coefficients. The asterisk marks the point where the RMSE_CV_ reaches its minimum value and indicates the corresponding number of variables, representing the optimal variable subset. In the VNIR spectral range, the number of feature variables decreased from 646 to 33, and in the SWIR spectral range, they were reduced from 148 to 19, respectively. It effectively reduced the number of feature variables while retaining valuable information. The corresponding wavelength data are provided in Table 2.

The wavelenghths of the feature variables selected in this study are consistent with those reported in the references [34,35]. In the VIS region, the feature wavelengths are primarily concentrated within the ranges of 520–540 nm, 560–600 nm, and 630–700 nm. These spectral bands reflect both the intrinsic chromophores of tannin molecules and the color effects resulting from their interactions with metal ions, which lead to visible differences in sorghum grain coloration. In the NIR region, the feature wavelengths are mainly concentrated near 830–860 nm and 885–900 nm, likely associated with the stretching vibrations of C–H and C–O bonds within the grains. In the SWIR region, wavelengths such as 1048, 1129, 1133, 1289, 1327, 1350, 1383, 1572, 1595, and 1633 nm may be associated with C–H bond vibrations; bands at 1180, 1402, 1416, and 1520 nm are likely related to O–H bonds. These indicators characterize the overall biochemical environment of tannin and are indirectly correlated with tannin content. Wavelengths such as 1430, 1473, 1482, 1553, and 1558 nm may correspond to N–H bond vibrations, and are a direct indicator of tannin-protein complexes and strongly associated with tannin content. These feature variables effectively capture biochemical signals that co-vary with tannin content, demonstrating the advantage of hyperspectral technology in revealing core compositional indicators through the correlation of spectral features.

The PLS prediction models using whole variables and CARS variables were established, as shown in Table 3. Based on various evaluation metrics, the performance of the CARS-PLS model surpassed that of the whole-PLS model. The R_C_^2^ values of the CARS model for VNIR, SWIR and VNIR-SWIR increased by 0.03, 0.05, and 0.04, the RMSE_C_ decreased by 0.03, 0.02, and 0.03, and the RPD_C_ increased by 0.18, 0.08, and 0.20, respectively. These results indicate that the CARS algorithm effectively reduces dimensionality with preserved signal integrity [36,37]. This process not only reduces model complexity but also enhances prediction accuracy.

### 3.4. Comparison of Prediction Models and Optimal Prediction Model

#### 3.4.1. Comparison of Prediction Models

In this study, PLS, SVM, and CNN prediction models were first established using raw spectral data subset from VNIR-Raw, SWIR-Raw, and a direct concatenation of VNIR and SWIR (Whole-Raw), respectively. Subsequently, three predictive models were established using the VNIR-Feature, SWIR-Feature, and Fused-Feature subsets, respectively. The prediction results of all models are presented in Figure 5, Figure 6 and Figure 7.

The model prediction results demonstrate that the raw spectrum model constructed by directly concatenating data from VNIR and SWIR achieved higher predictive accuracy compared to models based on either sensor alone. Taking the SVM prediction model as an example, the R_P_^2^ of the concatenated Whole-Raw model reached 0.80, representing an improvement of 7.20% and 48.75% over the VNIR and SWIR models, respectively. The RMSE_P_ was 0.33, which decreased by 6.14% and 33.92% compared to the VNIR and SWIR models, respectively. Additionally, the RPD_P_ reached 2.24, showing an increase of 12.75% and 52.08% over the VNIR and SWIR models, respectively. Consistent trends were also observed in the PLS and CNN models, confirming that the data concatenation strategy using dual hyperspectral sensors enhances model prediction performance.

To further enhance model performance, this study employed the CARS method to extract characteristic bands from VNIR and SWIR data, constructing VNIR-Feature and SWIR-Feature subsets, and subsequently built a fused-feature subset through feature fusion technology. Predictive models were then developed using PLS, SVM, and CNN algorithms. The results demonstrated that the models built with each feature subset outperformed the full-spectrum model in predictive accuracy. Taking the SVM model as an example, the R_P_^2^ for the VNIR-Feature, SWIR-Feature, and Fused-Feature models were 0.78, 0.58, and 0.82, respectively, representing improvements of 4.06%, 7.98%, and 2.78% over the raw spectral models. The RMSE_P_ values were 0.34, 0.48, and 0.30, corresponding to reductions of 3.90%, 4.76%, and 10.79%, respectively. The RPD_P_ values were 2.14, 1.55, and 2.38, showing increases of 7.69%, 5.11%, and 11.25%, respectively. Notably, the model built with the Fused-Feature subset comprehensively outperformed those based on single-sensor feature subsets. Again using the SVM model as an example, the Fused-Feature model showed R_P_^2^ improvements of 5.87% and 41.58% over the VNIR-Feature and SWIR-Feature models, respectively, with RMSE_P_ reductions of 12.86% and 38.17%, and RPD_P_ increases of 11.25% and 53.75%. The PLS and CNN models exhibited the same trend, indicating that the spectral feature fusion strategy further enhances the predictive capability of the models.

To statistically validate the observed performance differences, a one-way ANOVA followed by Tukey’s HSD post-hoc test was conducted on the RMSEP values. The one-way ANOVA results indicated a statistically significant difference in RMSEP among the different models (F(2, 27) = 115.74, *p* < 0.001). Post hoc multiple comparisons revealed that the predictive performance of both the SVM model and CNN model was significantly superior to that of the PLS model (all *p* < 0.001). Although the difference between the SVM model and CNN model did not reach statistical significance, the CNN model’s lower mean RMSEP suggests it may hold greater potential for practical application.

#### 3.4.2. Optimal Prediction Model

Figure 8 illustrates the fitting performance of the predictive models developed by integrating the Fused-Feature subset with the PLS, SVM, and CNN algorithms. As shown, after feature fusion, both the SVM and CNN models achieved coefficients of determination exceeding 0.80 in the prediction set, while the PLS model fell slightly below this threshold. Furthermore, all models exhibited RPDP values greater than 2.0 and maintained low RMSEP levels, indicating robust predictive capability across the three modeling approaches. Among them, the CNN model demonstrated the best overall performance metrics, with the optimal model achieving a R_P_^2^ of 0.83, a RMSE_P_ of 0.29, and a RPD_P_ of 2.42. The model’s performance meets the benchmark for practical application in food quality prediction. With an R_P_^2^ of 0.83, it demonstrates the “excellent predictive accuracy” suitable for quantitative analysis as per Nicolaï et al. [27]. Simultaneously, its RPD_P_ of 2.42 signifies a “robust model” reliable for screening, according to Williams & Sobering [28]. The predictive performance was further evaluated by calculating the Bias, Standard Error of Prediction (SEP). A Bias of −0.0039 indicated a slight underestimation tendency in the model. The SEP of 0.2893 demonstrated the precision and reproducibility of the predictions around this systematic error. The 95% confidence interval for a single new prediction was between 1.1412 and 1.2907. Residual analysis, presented in Figure 9, demonstrates that the prediction errors of the CNN model approximately follow a normal distribution with no evident systematic bias.

## 4. Discussion

### 4.1. Discussion of Sample Representativeness and Dataset Reliability

This study collected 240 sorghum samples from six regions in Shanxi Province. These samples exhibited substantial variation in phenotypic traits, including cultivar, grain color, size, and shape, thereby ensuring the diversity of the sample set. The detection model developed from this representative dataset successfully captured generalizable relationships between tannin content and spectral features. This capability enables accurate prediction for unknown sorghum samples from different geographical origins or cultivars, demonstrating strong generalization performance [38].

The tannin content in the samples was determined in strict accordance with the national standard using spectrophotometry. This method provided precise and reliable reference values, effectively avoiding systematic errors introduced by chemical measurement deviations, thereby laying a solid foundation for establishing high-precision hyperspectral quantitative analysis models.

### 4.2. Discussion of Dual Hyperspectral Data Sources and Feature Fusion Strategy

#### 4.2.1. Complementarity of Dual Hyperspectral Data

Based on the aforementioned sorghum sample set, this study investigated the effectiveness of dual hyperspectral data fusion. The results demonstrate that the Whole-Raw data model outperformed any single data model. This can be attributed to the informational complementarity between different spectral regions. The VNIR region primarily captures information related to chromophores and electronic transitions, whereas the SWIR region is more sensitive to combination and overtone vibrations of molecular bonds such as O–H, C–H, and N–H. Differences in physical traits and tannin content among various sorghum varieties may thus be distinctly captured by different sensors. Thus, the fusion of dual hyperspectral data provides a more comprehensive information foundation, highlighting the complementary nature of multi-source information.

#### 4.2.2. The Impact of Feature Extraction and Fusion on Model Performance

The application of the CARS feature extraction method reduced the dimensionality of the data. In the VNIR spectral range, the number of feature variables decreased from 646 to 33, a reduction of 94.89%, and in the SWIR spectral range, they were reduced from 148 to 19, a reduction of 87.16%. The performance of the feature models over the raw spectrum models indicates that the issues of collinearity and spectral redundancy inherent in hyperspectral data have been effectively mitigated. The CARS algorithm employs a competitive adaptive reweighted sampling mechanism to select the most relevant and information-rich feature bands associated with tannin content. By eliminating redundant information and noise, the model structure is simplified, thereby enhancing both the robustness and generalization capability of the models [39].

Furthermore, the superior performance of the CARS-Fused-Feature model over both the CARS-VNIR and CARS-SWIR models indicates that the feature fusion strategy successfully extracts key information from the two sensor datasets and integrates them organically [40]. This strategy significantly enhances the density of effective information, enabling the modeling algorithm to focus more on decisive features, thereby achieving optimal predictive performance.

In summary, the feature extraction and feature fusion approach constitutes a more efficient and superior data processing workflow, which enhances prediction performance while simultaneously streamlining the model.

### 4.3. Discussion on Predictive Model Performance

#### 4.3.1. Comparative Analysis of Linear Versus Nonlinear Models

Among the predictive models developed, the linear PLS model demonstrated inferior overall performance compared to the nonlinear SVM and CNN models. This discrepancy primarily stems from the models’ differing capabilities in adapting to data characteristics and structure [41,42]. This study directly employed raw spectral data without preprocessing steps such as smoothing or scatter correction, aiming to preserve complete information in the original spectra. However, the relationship between sorghum tannin content and spectral features is likely influenced by multiple factors, including scattering effects and inter-component interactions, thereby exhibiting pronounced nonlinear characteristics that exceed the capacity of linear models. As a linear model, PLS struggles to capture the complex nonlinear characteristics present in raw spectral data. In contrast, SVM handles nonlinear problems by employing kernel functions to map the original data into a higher-dimensional space where the relationship becomes linear [43], whereas CNN leverages its architecture of multiple nonlinear activation layers and local receptive fields to automatically learn deep-level features within spectral data [28,44]. Thus, both SVM and CNN demonstrate superior adaptability when modeling such complex, unprocessed spectral data, enabling them to more accurately fit the true mapping relationship between tannin content and spectral response.

#### 4.3.2. Comparative Analysis of SVM Versus CNN

Among the nonlinear models, the CNN model achieved superior performance, while the SVM model performed slightly less effectively. This difference arises because SVM, as a shallow architecture, is critically dependent on the selection of kernel functions and parameters. In contrast, CNN possesses the capability for automated deep feature learning, enabling it to progressively extract deeper correlations through successive convolution and pooling operations applied to the selected features, thereby uncovering subtle patterns that are challenging for SVM to detect [45,46].

In summary, based on a high-quality sorghum sample set encompassing diverse genetic and environmental backgrounds, this study systematically demonstrated the significant advantages of multi-sensor feature fusion strategy and nonlinear deep learning models in hyperspectral quantitative inversion of tannin content. The proposed method provides a reliable and practical technical solution for rapid, non-destructive quality assessment of tannin content of sorghum.

### 4.4. Industrial Application, Techno-Economic Assessment and Operational Considerations

According to the newly implemented national standard for sorghum (GB/T 8231-2024), which took effect in 2024, there is no unified requirement for tannin content in sorghum used for liquor brewing [47]. However, within the diverse flavor system of Chinese liquor, tannin content still plays a critical role in shaping different aroma types. For instance, to ensure the formation of the sauce-aroma, Hongyingzi Company strictly controls the tannin content of its raw sorghum within the range of 1.0–2.0% [48]. In the production of strong-aroma liquor, a tannin content of approximately 1% contributes to a cleaner and more harmonious liquor body. In the brewing of light-aroma liquor, tannin levels exceeding 1.4% may inhibit yeast growth and metabolism, thereby affecting fermentation yield. The CNN model for sorghum tannin quantification enables direct integration with process analytical technology platforms, allowing for real-time, non-destructive detection of tannin content in incoming sorghum batches. This capability not only ensures that raw material quality meets enterprise preset standards but also streamlines the raw material acceptance process, thereby providing robust support for precise control and standardized management throughout the brewing process.

A preliminary techno-economic assessment underscores the feasibility of this approach. The primary capital investment would involve a hyperspectral imaging system and an embedded computing unit for model inference. Contrasted with the recurring costs and time delays associated with conventional laboratory wet chemistry, the proposed method offers substantial long-term savings. These savings are realized through high-throughput analysis, non-destructive testing, and the prevention of production bottlenecks caused by waiting for lab results. The rapid prediction capability of our optimized model makes it suitable for integration into a conveyor belt system to inspection all raw materials.

Several operationalization factors are critical for successful implementation. Firstly, the model would need to be deployed on robust industrial computing hardware. Secondly, personnel training for system operation and basic maintenance is essential. Finally, a seamless data interface with the plant’s central control system would be required to automatically accept or flag incoming sorghum batches based on the model’s prediction, thereby closing the loop on quality control and enhancing overall operational efficiency.

## 5. Conclusions

This study successfully established a non-destructive method for predicting sorghum tannin content by fusing data from VNIR and SWIR hyperspectral sensors. The key finding is that a feature-level fusion strategy, which integrates complementary spectral information from both sensors, coupled with the nonlinear modeling power of a CNN algorithm, yielded the optimal prediction performance (R_P_^2^ = 0.83, RMSE_P_ = 0.29, RPD_P_ = 2.42). This approach significantly outperformed models based on single-sensor data or linear methods. For practical implementation in line quality control, the developed model presents a viable path toward automated, real-time tannin detection in sorghum.

Looking forward, a deployment roadmap for online sorting is proposed. Model transfer to an industrial line would first require calibration transfer algorithms to minimize performance drift between the lab and target equipment. Sensor integration must address the synchronization of VNIR and SWIR cameras on a fast-moving conveyor belt. To maintain accuracy, a calibration update protocol is essential, using periodically sampled and chemically analyzed sorghum to recalibrate the model against source variations. Finally, rigorous inline validation would be conducted by correlating the predicted tannin values with key brewing metrics, such as liquor yield and flavor profiles, over extended production runs. These steps can collectively translate our detection model into a functional industrial technology, ultimately aimed at standardizing raw materials and enhancing final product quality.

## Figures and Tables

**Figure 1 foods-14-03880-f001:**
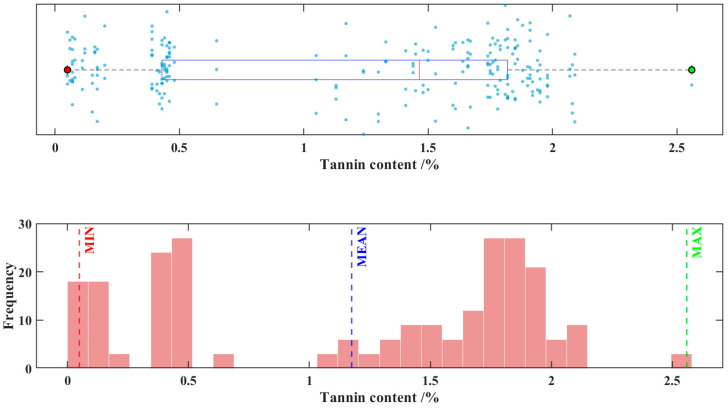
Scatter plot and frequency distribution histogram of tannin content in 240 sorghum samples.

**Figure 2 foods-14-03880-f002:**
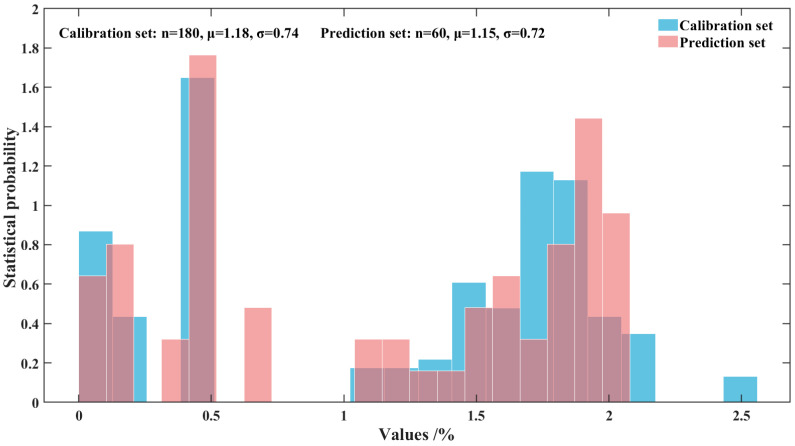
Histogram of sample distribution for the calibration set and prediction set.

**Figure 3 foods-14-03880-f003:**
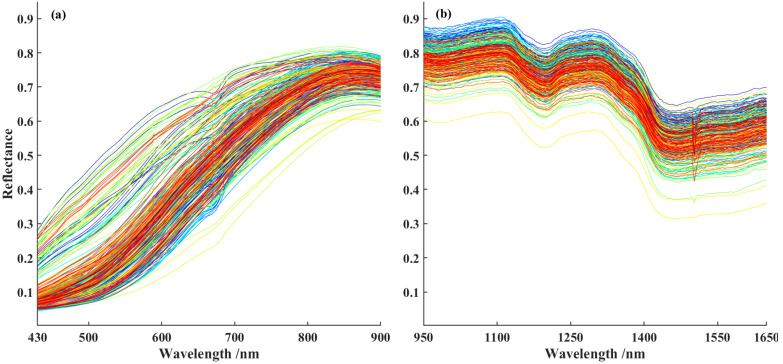
The average curve of raw spectra of 240 sorghum samples. (**a**) VNIR; (**b**) SWIR.

**Figure 4 foods-14-03880-f004:**
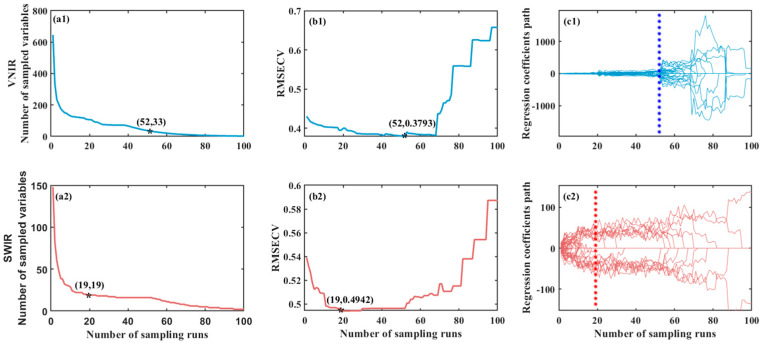
Extraction of key variables by CARS algorithm. (**a1**) The feature variables change of VNIR; (**b1**) The RMSECV values variation of VNIR; (**c1**) The variable regression coefficients trajectory of VNIR; (**a2**) The feature variables change of SWIR; (**b2**) The RMSECV values variation of SWIR; (**c2**) The variable regression coefficients trajectory of SWIR. And * represents the point where RMSECV reaches its minimum value.

**Figure 5 foods-14-03880-f005:**
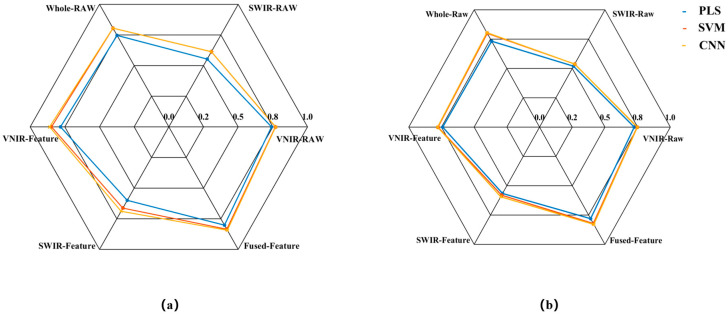
Radar chart for the R^2^ of the models. (**a**) R_C_^2^; (**b**) R_P_^2^.

**Figure 6 foods-14-03880-f006:**
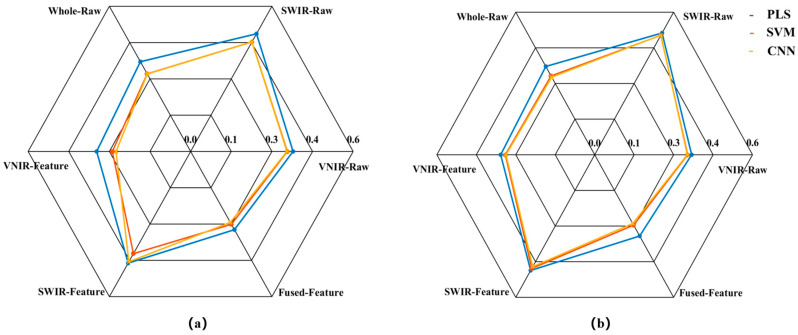
Radar chart for the RMSE of the models. (**a**) RMSE_C_; (**b**) RMSE_P_.

**Figure 7 foods-14-03880-f007:**
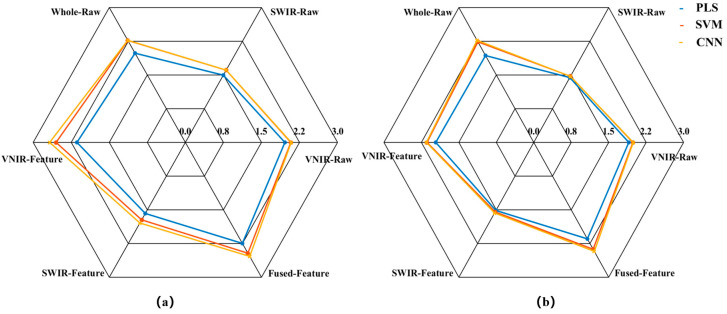
Radar chart for the RPD of the models. (**a**) RPD_C_; (**b**) RPD_P_.

**Figure 8 foods-14-03880-f008:**
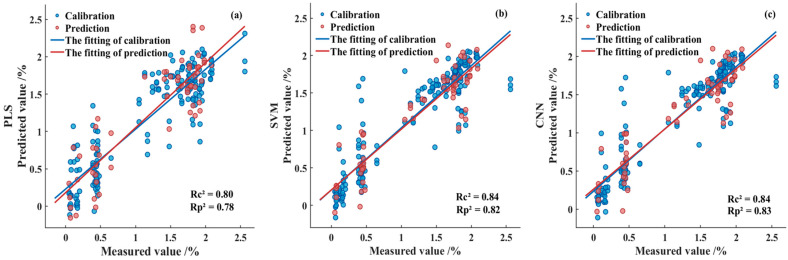
Fitting results of the calibration set and prediction set in the Fused-Feature prediction model. (**a**) PLS model; (**b**) SVM model; (**c**) CNN model.

**Figure 9 foods-14-03880-f009:**
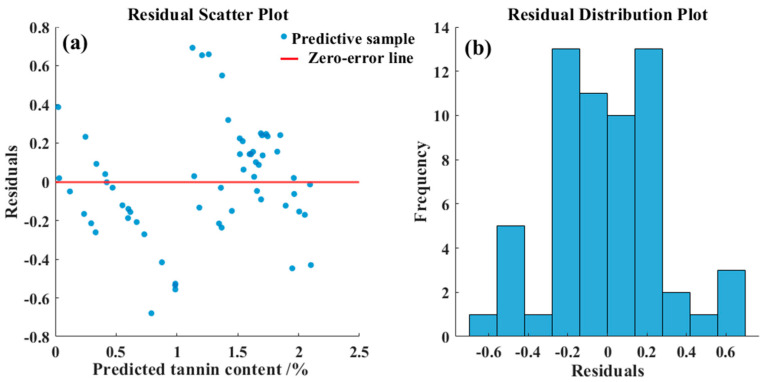
Residual analysis result. (**a**) Residual scatter plot; (**b**) Residual distribution plot.

**Table 1 foods-14-03880-t001:** The results of tannin content (%) dataset partitioning.

Tannin Content	Calibration Set					Prediction Set				
	Mean	Max	Min	SD	CV	Mean	Max	Min	SD	CV
VINR	1.16	2.56	0.05	0.74	0.64	1.21	2.09	0.05	0.71	0.58
SWIR	1.15	2.56	0.05	0.73	0.64	1.26	2.56	0.05	0.75	0.59
VINR + SWIR	1.18	2.56	0.05	0.74	0.63	1.15	2.07	0.05	0.72	0.63

**Table 2 foods-14-03880-t002:** Wavelength of feature variables proposed by CARS algorithm.

Sensor	Wavelength of Feature Variable/nm									
VNIR	522.115	533.744	534.471	536.651	538.831	564.269	565.723	567.903	570.811	571.537
	573.718	594.795	595.522	596.249	597.702	599.156	602.063	629.681	630.408	632.589
	633.315	634.769	648.578	650.032	652.212	681.284	703.088	828.824	843.36	854.989
	858.623	885.515	893.509							
SWIR	1048.94	1129.09	1133.8	1180.95	1289.39	1327.11	1350.68	1383.68	1402.54	1416.69
	1430.83	1473.26	1482.69	1520.41	1553.41	1558.13	1572.27	1595.84	1633.56	

**Table 3 foods-14-03880-t003:** Prediction results of PLS model based on whole spectrum and feature variables.

Sensor	Method	Number of Variables	Calibration Set					
			R_C_^2^	RMSE_C_	RPD_C_	R_CV_^2^	RMSE_CV_	RPD_CV_
VINR	whole-PLS	646	0.74	0.38	1.96	0.68	0.42	1.77
	CARS-PLS	33	0.78	0.35	2.14	0.74	0.38	1.96
SWIR	whole-PLS	148	0.55	0.49	1.50	0.48	0.53	1.39
	CARS-PLS	19	0.60	0.46	1.58	0.54	0.49	1.47
VNIR-SWIR	Whole-PLS	794	0.75	0.37	1.98	0.69	0.41	1.81
	CARS-PLS	52	0.79	0.34	2.18	0.76	0.36	2.05

## Data Availability

All the data mentioned in the article are included within the article, further inquiries can be directed to the corresponding author.

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
