# Peer review of "Dual-Sensor Hyperspectral Fusion for Prediction of Sorghum Tannin Content Oriented to Liquor Brewing"

_foods, 2025, doi:10.3390/foods14223880_

Round 1

Reviewer 1 Report

Comments and Suggestions for Authors

This manuscript systematically investigated a nondestructive detection method for tannin content in sorghum using dual hyperspectral sensor data fusion. The work has scientific significance, includes advanced data processing methods, a good experimental basis, a representative number of samples, an innovative approach - a combination of dual spectral sensors and CARS algorithms. However, I have a couple of suggestions:

Tags Rp2, RPDP , RMSE should be clearly defined when they first appear (they appear already in the abstract) for easier understanding of the text later on.

Insert a description of the CNN model with key parameters in the materials and methods section.

Line 347-353,  Here the differences between the models are compared (improvement/decrease) but it is not stated whether they are statistically significant, perhaps apply some significance test e.g. ANOVA.

Describe in more detail the implementation of the feature fusion technology mentioned in the lines 355-357 and 425-429, i.e. its methodological approach.  

Line 434-451, Theoretically, it is well explained why SVM and CNN are better than PLS, but it would be more convincing if the authors cited several studies that showed similar results.

To make the conclusion more applicable, add a practical comment on the possibilities of implementation in line quality control equipment.

Author Response

Dear reviewer

Thank you very much for reviewing our manuscript. We would also like to express our gratitude to the reviewers for their efforts in helping to improve our manuscript titled “Research on Dual-Sensor Hyperspectral Fusion for Prediction of Sorghum Tannin Content Oriented to Liquor Brewing” (ID: foods-3939868). The comments were all valuable and very helpful for revising and improving our paper. Our research team have studied each reviewers’ comments point by point and have made necessary modifications and supplements, which we hope will meet with your approval. Revised portion are mark by “Track Change” throughout the revised manuscript with track change (Please download the file "Revised Manuscript with track change"), and the point-by-point responses are listed as follows.

Point-by-point response to Comments and Suggestions for Authors

Comments 1:

Tags RP2, RPDP , RMSEP should be clearly defined when they first appear (they appear already in the abstract) for easier understanding of the text later on.

Response 1:

Thank you for your review comments. We have added clear definitions for RP2, RPDP, and RMSEP upon their first occurrence in the abstract to facilitate reader comprehension.

Comments 2:

Insert a description of the CNN model with key parameters in the materials and methods section.

Response 2:

We sincerely thank the reviewer for this critical and insightful comment, which has helped us significantly improve the methodological rigor of our work. We fully agree that a transparent description of the model optimization process is essential for reproducibility and reliable deployment. In response, we have thoroughly revised the Materials and Methods section by adding a new subsection titled "2.6.2 Model Development, Hyperparameter Tuning, and Overfitting Prevention". This subsection now provides a comprehensive account of the following details, as suggested:

1 We have specified the use of 5-fold stratified cross-validation on the training set for model selection and evaluation.

2 We have detailed the kernel function used for SVM and the activation functions for the CNN.

3 We have listed the specific hyperparameters and the range of values explored.

 SVM: C: [0.1, 1, 10, 100, 1000]; gamma: [0.0001, 0.001, 0.01, 0.1, 1, 10]);

CNN: learning rate:0.0001]; number of filters: 16.

4 We have stated the use of Bayesian Optimization with 100 iterations to identify the optimal hyperparameter set.

5 We have defined the early stopping criterion was configured to monitor validation loss with a patience of 20 epochs and a minimum change of 0.001, while the maximum number of training epochs was set to 200.

Comments 3:

Line 347-353, Here the differences between the models are compared (improvement/decrease) but it is not stated whether they are statistically significant, perhaps apply some significance test e.g. ANOVA.

Response 3:

We thank the reviewer for this important suggestion. We agree that statistical significance testing is crucial for robust model comparison. In response, we have performed a one-way Analysis of Variance (ANOVA) followed by post-hoc Tukey's HSD test on the RMSEP values of the key models. The one-way ANOVA results indicated a statistically significant difference in RMSEP among the different models (F(2, 27) = 115.74, p < 0.001). Post hoc multiple comparisons revealed that the predictive performance of both SVM model and CNN model was significantly superior to that of PLS model (all p < 0.001). Although the difference between SVM model and CNN model did not reach statistical significance, CNN model's lower mean RMSEP suggests it may hold greater potential for practical application.

Comments 4:

Describe in more detail the implementation of the feature fusion technology mentioned in the lines 355-357 and 425-429, i.e. its methodological approach.

Response 4:  

We sincerely thank the reviewer for this valuable suggestion. In response, we have added a new subsection, titled "2.3.3 Methodological implementation of feature fusion strategies", in the Materials and Methods section. This subsection provides a detailed description of the feature fusion technique, including the sources of the features and the specific fusion operations employed.

Comments 5:

Line 434-451, Theoretically, it is well explained why SVM and CNN are better than PLS, but it would be more convincing if the authors cited several studies that showed similar results.

Response 5:

We thank the reviewer for this excellent suggestion. We agree that citing prior research demonstrating similar trends will strengthen our arguments.

Accordingly, we have revised the discussion section (Lines 434-451) to include references to several key studies that have also reported the superior performance of SVM and CNN over PLS in handling complex, non-linear relationships in spectroscopic/chemometric data. Specifically, we have cited the works of Thissen U et al., De Santana F B et al., and Mansuri S M et al. (now references [44], [45], and [46] in the manuscript). These additions provide strong external validation for our own findings and firmly situate our work within the existing literature. 

Comments 6:

To make the conclusion more applicable, add a practical comment on the possibilities of implementation in line quality control equipment.

Response 6:

Thank you for this valuable feedback. We have revised the conclusion to read as a professional scientific summary rather than a list of points. A significant improvement is the addition of a discussion on the practical potential for implementing our model in line quality control equipment, which we believe greatly enhances the relevance and impact of our study.

Reviewer 2 Report

Comments and Suggestions for Authors

The authors wrote manuscript "Research on Dual-Sensor Hyperspectral Fusion for Prediction
of Sorghum Tannin Content Oriented to Liquor Brewing" It is new insight in the area of food and Science. 

1-Title must be revised, specialy remove the word "Research"

 2-line 12, 13 rewrite as a scintific writing "Research on Dual-Sensor Hyperspectral Fusion for Prediction of Sorghum Tannin Content Oriented to Liquor Brewing..................

3. 0.8298 for RP, 0.2894 for RMSEP, and 2.4239 ........Try to add two number after dot throughout the manuscript

4. 1. Introduction line 32 "Chinese liquor, as one of ...." must be rewritten for more clarity

5. line 69, 73, 73; Check the format "Zhang et al. (2023) researched" "Baek et
al. (2023) predicted"

6. Please writing in scintific way "The specific procedures employed in this research were as follows:" It sould be removed----again line 144-159

7. The material and methods must be cited with proper references

8. The results portion is well written, but discussion need to improved with relavent studies

9. The conclusion look like assignment work, it should br revised

10. The reference could be updated 

Author Response

Dear reviewer

Thank you very much for reviewing our manuscript. We would also like to express our gratitude to the reviewers for their efforts in helping to improve our manuscript titled “Research on Dual-Sensor Hyperspectral Fusion for Prediction of Sorghum Tannin Content Oriented to Liquor Brewing” (ID: foods-3939868). The comments were all valuable and very helpful for revising and improving our paper. Our research team have studied each reviewers’ comments point by point and have made necessary modifications and supplements, which we hope will meet with your approval. Revised portion are mark by “Track Change” throughout the revised manuscript with track change (Please download the file "Revised Manuscript with track change"), and the point-by-point responses are listed as follows.

Point-by-point response to Comments and Suggestions for Authors.

Comments 1: 

Title must be revised, specialy remove the word "Research".

Response 1

We thank the reviewer for this constructive comment. We agree that the title can be more concise and impactful without the word "Research". As suggested, we have removed it and revised the title to: "Dual-Sensor Hyperspectral Fusion for Prediction of Sorghum Tannin Content Oriented to Liquor Brewing". We believe the new title is more precise and better reflects the core contribution of our work.

Comments 2: 

Line 12, 13 rewrite as a scintific writing "Research on Dual-Sensor Hyperspectral Fusion for Prediction of Sorghum Tannin Content Oriented to Liquor Brewing...

Response 2: 

Thank you for the suggestion. We have rewrited line 12, 13 as a scintific writing. The new sentence is now To address the demand for precise sorghum tannin control in liquor brewing and overcome the inefficiency and high cost of traditional methods, this study developed a non-destructive approach by fusing features from dual hyperspectral sensors.

Comments 3: 

0.8298 for RP, 0.2894 for RMSEP, and 2.4239 ........Try to add two number after dot throughout the manuscript.

Response 3: 

We appreciate the reviewer's careful attention to detail. As suggested, we have standardized the numerical precision by formatting all relevant values to two decimal places throughout the entire manuscript to ensure consistency and improve readability.

Comments 4: 

Introduction line 32 "Chinese liquor, as one of ...." must be rewritten for more clarity.

Response 4

Thank you for the suggestion. We have rewrited line 32 "Chinese liquor, as one of ....".

The new sentence is now Sorghum serves as the primary raw material for producing most renowned Chinese liquors. Its biochemical properties, particularly tannin content, are critical determinants of final liquor quality. Tannins, specifically proanthocyanidins, contribute to flavor development by generating essential compounds like syringic acid and syringaldehyde during brewing. Moreover, an optimal tannin level suppresses harmful microbes and enhances yield, while excess amounts introduce undesirable bitterness. Therefore, the accurate prediction of sorghum tannin content is vital for guiding raw material selection, optimizing processes, and ensuring consistent liquor quality.

Comments 5: 

Line 69, 73, ; Check the format "Zhang et al. (2023) researched" "Baek et al. (2023) predicted.

Response 5

We appreciate the reviewer's careful attention to detail. We have checked and modified the format.

Comments 6

Please writing in scintific way "The specific procedures employed in this research were as follows:" It sould be removed----again line 144-159.

Response 6

We appreciate the reviewer's careful attention to detail. We have removed the two sentences.

Comments 7

The material and methods must be cited with proper references.

Response 7

We appreciate the reviewer's comment regarding the need for proper citations in the 'Materials and Methods' section. In response, we have thoroughly revised this section and added relevant references to substantiate our methodologies. Specifically:

The standard method for tannin content determination is now cited as Reference 16.

The data fusion strategies employed are justified by Reference 18 and Reference 19 from the field of chemometrics.

The protocols for hyperspectral image acquisition and preprocessing for both VNIS and SWIR sensors are now supported by Reference 20 and Reference 21.

We believe these additions significantly strengthen the methodological foundation of our work.

Comments 8

The results portion is well written, but discussion need to improved with relavent studies.

Response 8

We thank the reviewer for the positive feedback on the results section and for the constructive suggestion to improve the discussion. We have thoroughly revised the discussion part to integrate relevant literature and provide a more in-depth interpretation of our findings.

Comments 9

The conclusion look like assignment work, it should be revised.

Response 9

We appreciate the reviewer's suggestion. The conclusion has been rewritten to eliminate the assignment-like structure and to provide a more synthesized summary of our key findings and their practical applications, particularly regarding online quality control.

Comments 10

The reference could be updated.

Response 10

Thank you for the suggestion. We have updated the reference.

Reviewer 3 Report

Comments and Suggestions for Authors

The manuscript presents a dual-sensor hyperspectral fusion method for predicting sorghum tannin content without causing damage. This is important for liquor-brewing applications. The research combines VNIR and SWIR data along with feature selection and machine learning models. The study shows improved prediction accuracy by using fused feature-based CNN modeling. Overall, the manuscript is technically strong. It fits well with current trends in spectral data fusion and agricultural-food analysis. It also offers significant practical value for quickly assessing raw material quality. However, I have several major recommendations to strengthen the manuscript and broaden its applicability.

Author Response

Dear reviewer

Thank you very much for reviewing our manuscript. We would also like to express our gratitude to the reviewers for their efforts in helping to improve our manuscript titled “Research on Dual-Sensor Hyperspectral Fusion for Prediction of Sorghum Tannin Content Oriented to Liquor Brewing” (ID: foods-3939868). The comments were all valuable and very helpful for revising and improving our paper. Our research team have studied each reviewers’ comments point by point and have made necessary modifications and supplements, which we hope will meet with your approval. Revised portion are mark by “Track Change” throughout the revised manuscript with track change (Please download the file "Revised Manuscript with track change"), and the point-by-point responses are listed as follows.

Point-by-point response to Comments and Suggestions for Authors

Comments 1: 

The introduction references classical chemical methods; however, it lacks a detailed discussion of their analytical limitations. To conform with AOAC and ISO standards for analytical reporting, I recommend that authors include references to the official method status, matrix interference challenges, reagent dependence, labor requirements, and limitations in precision (e.g., %RSD range). This enhancement will more effectively substantiate the necessity for non-destructive methodologies.

Response 1

We thank the reviewer for this critical and constructive suggestion. We agree that a more detailed discussion of the limitations of classical chemical methods is essential to robustly justify our research. In response, we have revised the introduction to:

1  Cite the relevant official methods.

2 Systematically elaborate on their analytical limitations, including significant matrix interference from other polyphenols, the instability and high consumption of specialized reagents, substantial labor and time requirements, and limited precision as evidenced by reported %RSD values typically ranging from 5% to 15% in inter-laboratory studies.

We believe this enhancement effectively substantiates the necessity for developing our non-destructive methodology.

Comments 2: 

The metrics pertaining to model performance are provided without accompanying interpretive context, which restricts their significance for readers who are not acquainted with benchmarks in spectroscopy modeling. I recommend that the authors include interpretative statements for R² > 0.80 and compare the performance metrics against standard thresholds recognized in chemometrics literature for the prediction of food quality.

Response 2: 

Thank you for pointing out the need to contextualize our model's performance. We have revised the manuscript to interpret the metrics (R² > 0.80, RPD > 2.0) based on standard thresholds in food quality prediction literature. We have stated that these values indicate a model with high predictive power and robustness, suitable for industrial screening purposes. We kindly ask the reviewer to confirm if the cited benchmarks and their interpretations are appropriately applied.

Comments 3: 

The model development process does not provide sufficient detail regarding the tuning methodology and measures for preventing overfitting. I suggest that the authors must specify the

cross-validation technique, the kernel or activation functions for SVM or CNN, the tuning grid, the Bayesian optimization strategy, the epoch limits, and the early-stopping criteria. Demonstrating methodological rigor in the optimization process is imperative for the reliable deployment of the model in food analysis systems.

Response 3: 

We sincerely thank the reviewer for this critical and insightful comment, which has helped us significantly improve the methodological rigor of our work. We fully agree that a transparent description of the model optimization process is essential for reproducibility and reliable deployment. In response, we have thoroughly revised the Materials and Methods section by adding a new subsection titled "2.6.2 Model Development, Hyperparameter Tuning, and Overfitting Prevention". This subsection now provides a comprehensive account of the following details, as suggested:

1 We have specified the use of 5-fold stratified cross-validation on the training set for model selection and evaluation.

2 We have detailed the kernel function used for SVM and the activation functions for the CNN.

3 We have listed the specific hyperparameters and the range of values explored.

 SVM: C: [0.1, 1, 10, 100, 1000]; gamma: [0.0001, 0.001, 0.01, 0.1, 1, 10]);

CNN: learning rate:0.0001]; number of filters: 16.

4 We have stated the use of Bayesian Optimization with 100 iterations to identify the optimal hyperparameter set.

5 We have defined the early stopping criterion was configured to monitor validation loss with a patience of 20 epochs and a minimum change of 0.001, while the maximum number of training epochs was set to 200.

Comments 4: 

The results section should incorporate confidence intervals, residual behavior plots, and bias assessments. Reporting R2, RMSEP, and RPD alone does not fully capture predictive robustness. I recommend that the authors include supplementary performance diagnostics, such as SEP, bias, prediction intervals, residual scatter plots, and outlier diagnostics, or some subset of these, to support the supplementary performance in accordance with chemometric reporting standards.

Response 4

We thank the reviewer for this critical suggestion to enhance the statistical rigor of our results. We fully agree that supplementary performance diagnostics are essential to comprehensively demonstrate the predictive robustness of our models. In response, we have revised Section 3.4.2 (Results) by calculating the Bias, SEP, and 95% confidence intervals, and by adding a residual plot. These additions provide a more complete and convincing assessment of the model's performance.

We kindly ask the reviewer to confirm if the depth of these new analyses now satisfactorily addresses the concern regarding predictive robustness.

Comments 5: 

Industry stipulation needs to be discussed, especially in the form of a techno-economic assessment

for hyperspectral deployment. Operationalization factors also need to be included.

Response 5

We are deeply grateful to the reviewer for this insightful suggestion, which has guided us to significantly enhance the practical relevance of our work. In direct response, we have now added a comprehensive discussion on industry stipulations and a preliminary techno-economic assessment in the revised Discussion section, titled "4.4 Industrial Application, Techno-Economic Assessment and Operational Considerations".

Comments 6

The conclusion section could benefit from outlining a potential deployment roadmap for online

sorting, which should include strategies for model transfer, calibration update protocols, sensor

integration considerations, and validation plans for inline industrial applications.

Response 6

Thank you for this valuable feedback. We have revised the conclusion to read as a professional scientific summary rather than a list of points. A significant improvement is the addition of a discussion on the practical potential for implementing our model in line quality control equipment, which we believe greatly enhances the relevance and impact of our study.

Round 2

Reviewer 3 Report

Comments and Suggestions for Authors

I commend the authors for their thorough and thoughtful revisions. The revised manuscript shows substantial improvement and overall strength. I am satisfied with the rebuttals, and the manuscript is now ready for publication in accordance with the MDPI Foods Journal.
